# Metal Ion Binding to Human Glutaminyl Cyclase: A Structural Perspective

**DOI:** 10.3390/ijms25158279

**Published:** 2024-07-29

**Authors:** Giusy Tassone, Cecilia Pozzi, Stefano Mangani

**Affiliations:** 1Department of Biotechnology, Chemistry and Pharmacy, Department of Excellence 2018–2022, University of Siena, Via Aldo Moro 2, I-53100 Siena, Italy; giusy.tassone@unisi.it; 2Consorzio Interuniversitario Risonanze Magnetiche di Metallo Proteine (CIRMMP), Via Luigi Sacconi 6, I-50019 Sesto Fiorentino, Italy

**Keywords:** human glutaminyl cyclase, hQC, acyltransferase, X-ray crystallography, metal ions

## Abstract

Glutaminyl-peptide cyclotransferases (QCs) convert the N-terminal glutamine or glutamate residues of protein and peptide substrates into pyroglutamate (pE) by releasing ammonia or a water molecule. The N-terminal pE modification protects peptides/proteins against proteolytic degradation by amino- or exopeptidases, increasing their stability. Mammalian QC is abundant in the brain and a large amount of evidence indicates that pE peptides are involved in the onset of neural human pathologies such as Alzheimer’s and Huntington’s disease and synucleinopathies. Hence, human QC (hQC) has become an intensively studied target for drug development against these diseases. Soon after its characterization, hQC was identified as a Zn-dependent enzyme, but a partial restoration of the enzyme activity in the presence of the Co(II) ion was also reported, suggesting a possible role of this metal ion in catalysis. The present work aims to investigate the structure of demetallated hQC and of the reconstituted enzyme with Zn(II) and Co(II) and their behavior in the presence of known inhibitors. Furthermore, our structural determinations provide a possible explanation for the presence of the mononuclear metal binding site of hQC, despite the presence of the same conserved metal binding motifs present in distantly related dinuclear aminopeptidase enzymes.

## 1. Introduction

Glutaminyl-peptide cyclotransferases (QCs or QPCTs, EC 2.3.2.5) catalyze a post-translational chemical reaction in proteins or peptides. This reaction involves the conversion of N-terminal glutamine or glutamate residues to N-terminal pyroglutamate (pE), a process that produces the release of ammonia or a water molecule, respectively (Figure 1) [1,2,3]. The N-terminal pE modification protects the peptide/protein against proteolytic degradation by amino- or exopeptidases and increases protein stability [4]. 

**Scheme 1 ijms-25-08279-sch001:**
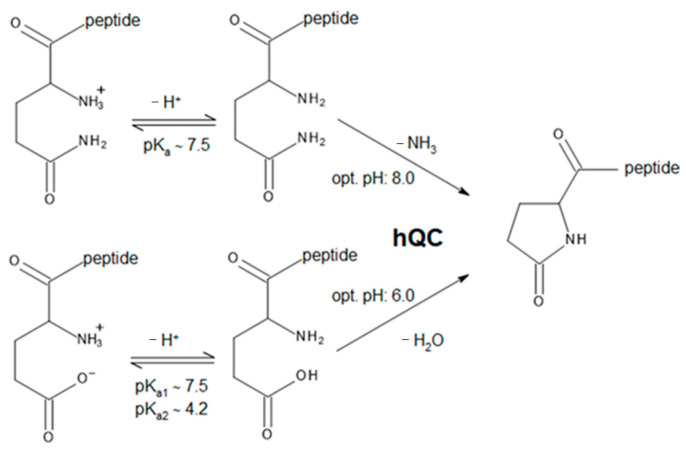
Schematic reactions catalyzed by hQC. Data regarding the optimum pH for the catalyzed reactions are from Schilling et al. [5].

The QC enzyme was first discovered in *Carica papaya* [6,7] and then in other eukaryotes and prokaryotes [8,9]. Two isoforms of QC are found in humans: the secreted hQC (or sQC) and the Golgi-resident enzyme (gQC) [10,11]. They have a similar size and a sequence similarity of 45% with the main difference in the N-terminus, which is either a nonpermanent signal for secretion or anchors the enzyme to the Golgi apparatus [11]. Mammalian QC is abundant in regions of the brain such as the hypothalamus and pituitary gland [12,13]. A large amount of evidence indicates that pE peptides are involved in the onset of neural human pathologies such as Alzheimer’s disease [14,15], Huntington’s disease [16,17,18,19], and synucleinopathies [20]. Notably, hQC plays a crucial role in the generation of pGlu-containing amyloid beta 3-40/42 peptides (pGlu-Aβ_3-40/42_), which are the main constituents of intracellular, extracellular, and vascular Aβ deposits in Alzheimer’s disease (AD) brains [21,22,23]. This enzyme also exerts a similar effect in Huntington’s disease (HD), where the mutant huntingtin protein, characterized by expanded polyglutamine regions, undergoes cleavage processes that lead to the accumulation of hQC-mediated pGlu protein aggregates [18]. These pGlu-modified peptides exhibit hydrophobicity, rapid aggregation, neurotoxicity, and resistance to degradation by aminopeptidases, thus accelerating the progression of these neurodegenerative disorders [24,25,26]. The expression of hQC is elevated in AD, HD, and other neurodegenerative disorders, highlighting the potential therapeutic benefit of inhibiting this enzyme to mitigate the formation of pGlu-modified deposits and plaques [18,21,27,28]. Hence, hQC has become an intensively studied target for drug development against these diseases [29,30,31,32]. Leaving aside the health-related aspects of hQC, the present work focuses instead on the chemical behavior of hQC with respect to metal ion binding and on the influence of different metal ions on the inhibition of the enzyme. 

hQC is composed of 361 amino acids (GenBank-id: EAX00394.1; UniProt: Q16769), of which the first 28 constitute the secretory signal. The mature protein consists of 333 amino acids. In the absence of glycosylation, as is the case for the heterologous expression of hQC in *E. coli*, the molecular weight is ~38 kDa. When expressed in *Pichia pastoris* or mammals, this value is increased by up to about 4 kDa due to glycosylation at sites Asn49 and Asn296. These are located in a less conserved part of the sequence and do not seem to be of catalytic relevance, since deglycosylation has no consequences for activity [1,33]. 

Soon after its characterization, hQC was proposed to be a Zn enzyme [34]. Indeed, its crystal structure revealed that a Zn(II) ion bound in a tetrahedral coordination to Asp159, Glu202, His330, and to a water molecule [13,33,35,36]. This binding site corresponds to the Zn1 binding site of a large group of related aminopeptidases belonging to the clan M28.002 (MEROPS accession: MER0001284) [34,35,37]. The M28 clan enzymes share the same conserved residues as hQC that bind Zn1, as well as the geometry of the bound metal. Adjacent to the Zn1 binding site, the M28 aminopeptidases bind a second Zn(II) ion. Although the amino acids involved in binding Zn2 are completely conserved in hQC (His140, Asp159, Asp248), the crystal structures show that the Zn2 site of hQC is not occupied. Former studies on hQC also reported a partial restoration of the enzyme activity in the presence of the Co(II) ion, suggesting a possible role of this metal ion in catalysis [34]. Furthermore, it was suggested that Co(II) ions can compete with Zn(II) for enzyme binding, but the interaction of hQC with other metal ions was not further investigated. 

This work is focused on the bioinorganic chemistry of hQC, aiming to characterize, from the structural point of view, the metal binding capability of the enzyme and to understand why hQC has a mononuclear metal binding site in spite of the presence of residues able to host a second metal ion. Our investigations involve the crystal structures of the demetallated (apo-) enzyme, of the reconstituted hQC with Zn(II) and Co(II), and of the adducts of Co-hQC with the known inhibitors PBD-150 [36,38] and SEN-177 (Figure 1) [32]. All X-ray crystallography experiments were performed using the hQC double mutant Y115E-Y117E (hQC-2X), formerly validated by us as a soluble protein variant exploitable for drug discovery purposes [13,39].

## 2. Results and Discussion

### 2.1. Crystal Structure of the Apo-State of hQC-2X

The crystal structure of apo-hQC-2X was determined to be 1.96 Å resolution, in the monoclinic space group C2. The crystal asymmetric unit (ASU) contains three independent molecules that are almost identical within experimental errors (pairwise root-mean-square deviation, RMSD, values of 0.11–0.14 Å). The tertiary structure of hQC-2X, having the mixed α/β hydrolase fold with an open sandwich topology, is not affected by the removal of the functional metal cofactor. Indeed, the structural comparison between apo-hQC-2X and the holo-enzyme shows no meaningful difference, as indicated by RMDS values ranging from 0.21 Å to 0.24 Å. This evidence suggests that the binding of the metal cofactor, although critical for the catalytic function, is not necessary for the enzyme to acquire the proper folding.

The conformation of the catalytic cavity is well defined by the electron density, allowing all residues exposed in the active site of apo-hQC-2X to be modeled. Upon removal of the metal ion, a water molecule, WatC, populates the cofactor binding pocket in all three subunits composing the ASU (Figure 2). WatC donates two H-bonds to Asp159 and Glu202 and receives from His330 and a further water molecule (Wat2), placed inside the catalytic cavity and bound to Trp329 (Figure 2). On the opposite wall, the indole moiety of Trp207 is rotated inside the catalytic cavity, contributing to the cone-shaped active site (Figure 2). 

The superimposition of apo-hQC with either the native (PDB: 2AFM, 2AFO [35], and 4YU9 [36]) or reconstituted Zn(II)-hQC (structure reported below) results in overall RMSDs in the range of 0.43–0.45 Å, indicating the steadiness of the active site cavity residue orientation and preservation of the cavity shape. The main difference observed in the comparison between apo-hQC and the holo-enzyme (PDB id 4YU9 [36]) consists of the displacement of WatC by the Zn(II) ion, with Wat1 closing its tetrahedral coordination geometry (Figure 3). 

### 2.2. Structural Characterization of the Holo-States of hQC-2X with Zn(II) and Co(II) Metal Ions

To validate the protocol for obtaining metal derivatives starting from the apo-state of hQC-2X, the Zn(II) holo-enzyme was first generated. The structure of the hQC-2X in complex with Zn(II) ions, hQC-2X–Zn(II), was determined as a 1.78 Å resolution by crystallizing a sample of apo-protein exposed to 10 equivalents of Zn(II) ions (Figure 4A). The structure of hQC-2X–Zn(II) is superimposable to the formerly reported structural model of hQC-2X (PDB id 4YU9 [36]), showing the Zn(II) ion populating the metal cofactor pocket within the catalytic cavity (Figure 4A,B). The coordination of the metal ion is fully conserved in the two structural models, validating the metal derivatization protocol.

Despite the stoichiometric excess of Zn(II) ions used to obtain the holoenzyme, only the Zn1 site in the catalytic cavity hosts the metal ion. The putative Zn2 site present in bacterial aminopeptidases, defined by His140, Asp159, and Asp248, remains unoccupied.

The structure of the hQC-2X in complex with Co(II) ions, hQC-2X–Co(II) (Figure 4C,D) was determined to be 2.30 Å resolution by applying the same protocol. The structure of hQC-2X–Co(II) was obtained by crystallizing a sample of apo-protein exposed to 10 equivalents of Co(II) ions. The Co-derivative crystals are isomorphous to those of the Zn(II) derivative and of the apo-enzyme. All crystals belong to the monoclinic space group C2 with similar unit cell parameters, and the ASU is composed of three independent molecules. The protein fold is not significantly affected by the metal substitution; indeed, the overall fold of the Co(II) and Zn(II) derivatives is conserved, as indicated by the RMDS values ranging from 0.36 Å to 0.39 Å. The population of the metal cofactor site by Co(II) ions was verified by MAD, collecting data at energy above the cobalt K-edge (7750 eV) and immediately below the edge (7600 eV). A strong signal in the anomalous map computed with anomalous difference coefficients from the high-energy dataset was detected in the metal cofactor site, readily disappearing in the anomalous difference map calculated with the data obtained below the edge (Figure 4D). This provides evidence of Co(II) ions populating the metal cofactor pocket and confirms the effective generation of the hQC-2X–Co(II) derivative. The Co(II) ion is coordinated by the three protein residues, Asp159, Glu202, and His330, also involved in the coordination of Zn(II) ions in the hQC-2X–Zn(II) structure (Figure 4A,C). A water molecule, Wat1, is placed in the active site, 2.83–3.14 Å away from the Co(II) ion, forming a slightly distorted tetrahedron around the metal (Figure 4C). The position of Wat1 differs from that of the corresponding water molecule in the structure of hQC-2X–Zn(II) (Figure 4A). At variance with the Co(II) derivative, Wat1 is more tightly coordinated to the Zn(II) ion, showing a regular tetrahedral coordination geometry (Figure 4A,C). The conformation of the catalytic cavity in the structure of hQC-2X–Co(II) is well defined by the electron density, allowing all residues exposed in the active site to be modeled. As formerly observed for the apo-enzyme, the side chain of Trp207 is rotated inside the catalytic cavity, contributing to shaping the active site cone (Figure 4C,D). As observed for the reconstituted Zn-hQC, the putative second metal binding site (His140, Asp159, and Asp248) is empty.

### 2.3. Structural Characterization of hQC-2X–Co(II) in Complex with PBD-150 and SEN-177

The structures of hQC-2X–Co(II) in complex with PBD-150 and SEN-177 were determined to 3.06 Å resolution (Figure 5A and Figure 6A, and Table 1). The formation of the holo-enzyme with Co(II) ions was confirmed in both complexes by SAD data collection at energy above the cobalt K-edge (7800 eV). The presence of a Co(II) ion is indicated by the strong signal in the anomalous difference map calculated with the data obtained at energy above the cobalt edge (Figure 5B–E and Figure 6B–E). The Co(II) ion is coordinated to the enzyme active site by Asp159, Glu202, and His330, showing a conserved geometry with respect to the ligand-free state (Figure 4C, Figure 5A and Figure 6A). The only difference is the replacement of the water molecule completing the tetrahedral coordination geometry of the Co(II) ion in the ligand-free structure (Figure 4C), which is replaced by the exogenous ligands in both complexes’ states (Figure 5A and Figure 6A). The displacement of this water molecule also represents the first stage of the enzyme catalyzed reaction, allowing the binding of the substrate to the active site. Both inhibitors populate the catalytic cavity of all three enzyme subunits composing the crystal ASU, with conserved binding geometries and interactions. 

PBD-150 coordinates the metal ion through the terminal N atom of its imidazole ring (Figure 5A). The flexible carbon chain connecting the imidazole ring to the thioketone moiety points to the active site wall lined by Trp207, Ile303, Gln304, Phe325, and Trp329, stabilized by van der Waals interactions with Phe and Trp residues. The thiourea sulfur receives an H-bond from the backbone nitrogen of Gln304, and the thiourea nitrogen linked to the 3,4-dimethoxyphenyl ring forms water-mediated interactions with Tyr299 and Val302. These interactions favor the correct alignment of the dimethoxyphenyl moiety in the surface-exposed pocket lined by Tyr299, Ile303, Trp329, and the loop 323–325. The phenyl moieties of PBD-150 and Phe325 entail a parallel displaced π-π stacking interaction, whereas the *m*-methoxy oxygen receives an H-bond from the Phe325 backbone nitrogen (Figure 5A). The two methyl moieties point in opposite directions being stabilized by van der Waals interactions with Phe325 and Trp329.

Analogously to PBD-150, SEN-177 coordinates the cobalt ion through the nitrogen N1 of the methyl–triazole ring (Figure 6A). Its binding orientation places the following nitrogen N2~3.5 Å away from the indole nitrogen of Trp329, in a non-optimal H-bond interaction geometry. On the opposite side of the same ring, the methyl moiety in position 4 is accommodated inside the pocket lined by Asp243, Ile321, and the loop 303–305. The triazole ring of SEN-177 is functionalized in position 3 with a piperidine moiety linking it to the bipyridyl group. The piperidine ring is accommodated inside the hydrophobic channel lined by Trp207, Ile303, Gln304, Phe325, and Trp329, stabilized by van der Waals interactions with Phe and Trp residues. The first pyridine ring of the SEN-177 bipyridyl system is stacked over the Trp207 indole, whereas the second fluoro-pyridine moiety points towards Lys144 and His330. The aromatic ring is placed between Trp207 and Trp329; nearby, Lys144 and His330 donate H-bonds to the *p*-fluoro substituent. 

The binding modes of both inhibitors in the hQC-2X–Co(II) complexes are very similar to those formerly observed in the hQC-2X–Zn(II) complexes (Figure 7). The replacement of Zn(II) ions with Co(II) in the hQC active site does not alter the inhibitor binding; indeed, both metal ions are coordinated by the inhibitors with similar configurations.

### 2.4. Comparison of hQC with Bacterial di-zinc Aminopeptidases

It has been noticed that hQC shares aminopeptidases of the clan M28.002 with di-zinc, not only in terms of the residues and the geometry of the Zn1 binding site, but also in the conservation of the same amino acids that are responsible for binding the second zinc ion (Zn2) in M28 peptidases [25,26,28]. Figure 8A shows the superimposition of the zinc binding sites of the representative aminopeptidase from *Aeromonas proteolytica* (ApAP; PDB: 1AMP) with hQC-2X. It can be seen that all residues involved in the metal binding of both enzymes are almost coincident (positional differences between metal binding side chain atoms range: 0.50–1.34 Å). The structural reason for the missing Zn2 site in hQC should then be searched in the non-conserved amino acids present in the neighborhood of the active site. Figure 8A shows that in hQC, Ser160 and Pro163 replace Asp118 and Gly121 of ApAP, respectively. In ApAP, the Asp118 side chain receives a H-bond from the backbone N of Gly121 (3.09 Å). This H-bond has the effect of turning the side chain of Asp118 out of the active site leaving space for the binding of Zn2 (Figure 8A). The surface of the di-zinc site in ApAP is shown in Figure 8B. The hQC Pro163 has exactly the same position as ApAP Gly121 (Figure 8A), but the bulky side chain of Pro163 cannot establish an interaction with Ser160, whose side chain is rotated (both in Zn-bound and in the apo-form of hQC-2X) towards the active site by making a strong H-bond (2.60 Å, Figure 8A) to the side chain of Asp248 (corresponding to the ApAP Zn2 binding ligand Asp179). The hQC Ser160 OH falls at about 3.0 Å away from the putative Zn2 site, making the binding of a Zn(II) ion to the Zn2 site impossible. Figure 8C compares the shapes of the metal binding cavity of hQC-2X with that of ApAP (Figure 8B).

## 3. Conclusions

We have reported a protocol to demetallate and reconstitute metal-loaded hQC with both Zn(II) and Co(II) ions. The crystal structure of apo-hQC underlines the robustness of the hQC active site cavity that maintains its structure even without the metal ion. On the other hand, this scarcely flexible cavity indicates quite strict requirements for molecules that can be hosted as either substrates or inhibitors.

The hQC-2X–Co(II) and the complexes with PBD-150 and SEN-177 show that Co(II) ions can replace the catalytic Zn(II) and are able to bind molecules in the same pose as the native metal. This finding validates the possibility of using visible spectroscopy as well as electronic paramagnetic resonance (EPR) or paramagnetic NMR to study hQC in solution.

Structural comparisons of hQC-2X–Zn(II), hQC-2X–Co(II), and apo-hQC-2X with representative aminopeptitases of the M28 clan allow us to put forward the hypothesis that the missing Zn2 site in hQC is due to mutations occurring in the surroundings of the hQC metal binding site, namely, hQC Ser160 and Pro163 in place of ApAP Asp118 and Gly121. Studies for validating this hypothesis by producing hQC Ser160/Asp and Pro163/Gly variants in an attempt to insert a second Zn(II) ion are underway in our laboratory.

## 4. Materials and Methods

### 4.1. Protein Expression and Purification 

Recombinant hQC-2X was expressed and purified following established protocols [32,36], with minor modifications. Briefly, the protein was expressed in the *Escherichia coli* strain BL21(DE3) and purified by nickel affinity chromatography (HisTrap FF 5 mL column, GE Healthcare, Chicago, IL, USA) in a three-step gradient. The target protein was eluted by applying a 250 mM imidazole concentration in buffer A (150 mM NaCl and 20 mM Tris-HCl, pH 8). The purified fractions were extensively dialyzed in buffer A at 4 °C. The high purity (>98%) of the resulting sample was confirmed by SDS-PAGE analysis and MALDI-TOF mass spectrometry. The final production yield of hQC-2X was estimated to 82 mg L^−1^ of bacterial culture.

### 4.2. Preparation of Apo-hQC-2X and Co(II)-hQC-2X

Apo-hQC-2X was prepared from the purified protein by dialysis in buffer A supplemented with 5 mM EDTA (Merck KGaA, Darmstadt, Germany), as chelating agent, at 8 °C for 16 h. EDTA was then removed by extensive dialysis in buffer A at 8 °C. The sample of apo-hQC-2X was concentrated to 10 mg mL^−1^ (in buffer A) and stored at −20 °C until required for crystallization trials. 

The hQC-2X–Zn(II) holo-enzyme was prepared by incubating a sample of apo-hQC-2X (10 mg mL^−1^, in buffer A) with 10 equivalents of ZnCl_2_ for 30 min on ice. The Co(II)-substituted hQC-2X was prepared using a similar procedure. The sample of apo-hQC-2X (10 mg mL^−1^, in buffer A) was supplemented with 10 equivalents of CoCl_2_ (Merck KGaA, Darmstadt, Germany) and incubated on ice for 30 min. The resulting samples of hQC-2X – Zn(II) and hQC-2X–Co(II) were used for crystallization trials. 

### 4.3. Crystallization

The crystallization of apo-hQC-2X, hQC-2X–Zn(II), and hQC-2X–Co(II) was performed according to established procedures [36]. Briefly, crystals were grown using the sitting drop vapor diffusion technique [40] by mixing equal volumes of protein (5 mg mL^−1^ in 150 mM NaCl and 20 mM Tris-HCl, pH 8) and precipitant (0.1–0.2 M ammonium sulphate and 0.1 M MES, pH 6.0) solution, equilibrated over a 200 µL reservoir at 8 °C. The complexes of hQC-2X–Co(II) with SEN-177 and PBD-150 were obtained by the soaking procedure on very thin plate-shaped crystals of enzyme–metal complexes with 5 mM of each compound solubilized in 1,4-dioxane. Progressive crystal damage (crystal cracking and progressive dissolution) was macroscopically observed during the soaking procedure, allowing a maximal exposure time of 2 h to be reached. All crystals were transferred to cryoprotectant solutions, prepared by adding either 20% *v*/*v* glycerol (VWR, Milan, Italy) or 25% *v*/*v* ethylene glycol (Merck KGaA, Darmstadt, Germany) to the precipitants and then flash-freezing in liquid nitrogen.

### 4.4. Data Collection, Structure Solution, and Refinement

Diffraction data were collected at 100 K using synchrotron radiation in the Diamond Light Source (DLS, Didcot, UK) beamline I04 equipped with Eiger2 XE 16M detector. Data were integrated using XDS [41] and scaled with AIMLESS [42,43] from the CCP4 suite [44]. hQC-2X crystals are thin plates and this affected the quality of X-ray diffraction data, as evidenced by the R-values reported in Table 1. This is even more evident in the complexes with SEN-177 and PBD-150, where the crystal quality was further compromised by the soaking procedure. Furthermore, for collecting complete datasets without losing diffraction from these very thin plate crystals, we had to increase the rotation angle to 0.5°, keeping a very low exposure time, 0.03 s (Table 1). This data collection strategy was also responsible for the high R-values obtained in the reduction statistics. All crystals belonged to the monoclinic space group C2 with three protein chains in the asymmetric unit (Table 1). Structures were solved by the molecular replacement technique as implemented in the software MOLREP (v. 11.0 /22.07.2010) [45]. The structure of hQC-2X (PDB id 4YU9 [36], excluding non-protein atoms and water molecules) was used as a searching model. All structures were refined using REFMAC5 [46] from the CCP4 suite [44]. The molecular graphics software Coot (v. 0.9.6) [47] was used for visual inspection and manual rebuilding of the missing atoms in the electron density maps and to add solvent molecules. In the hQC-2X–Zn(II) structure, the binding of Zn(II) ions was confirmed by inspecting the anomalous difference maps computed from data collected above and below the metal K-edge (9720 eV and 9550 eV, respectively). In the hQC-2X–Co(II) structures, the location of the cobalt ion within the active site of all protein chains was confirmed by examining two anomalous difference Fourier maps. One map was generated using anomalous difference coefficients from data collected at energy just above the Co K-edge peak energy (7750–7800 eV), and the second one by using anomalous difference coefficients from data collected at energy below the Co K-edge (7600 eV). 

The binding of the PBD-150 and SEN-177 to hQC-2X was established by inspecting conventional Fourier difference maps that clearly evidenced the presence of the ligands inside the catalytic cavity. Both ligands were modeled according to the electron density. All maps were calculated by the FFT program [48] from the CCP4 suite. The final models were inspected manually and checked with Coot [47] and PROCHECK [49] and then validated through the Protein Data Bank (PDB) deposition tools. Structural figures were generated using the molecular graphic software CCP4mg (v.2.10.11) [50]. Data collection, processing, and refinement statistics are summarized in Table 1. Final coordinates and structure factors were deposited in the Protein Data Bank (PDB) under the codes 9FXG (apo-hQC-2X), 9FXH (hQC-2X–Co(II), 9FXI (hQC-2X–Co(II)–SEN177), and 9FXJ (hQC-2X–Co(II)–PBD-150).

## Data Availability

Crystal structure final coordinates and structure factors are available in the PDB (www.rcsb.org, accessed on 24 June 2024) under the codes 9FXG (apo-hQC-2X), 9FXH (-hQC-2X–Co(II), 9FXI (hQC-2X–Co(II)–SEN177), and 9FXJ (hQC-2X–Co(II)–PBD-150).

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
