# Peer review of "Metal Ion Binding to Human Glutaminyl Cyclase: A Structural Perspective"

_ijms, 2024, doi:10.3390/ijms25158279_

Round 1
Reviewer 1 Report
Comments and Suggestions for Authors
The manuscript by Tassone et al. describes the crystal structure determination of human glutamyl cyclase with different metals and inhibitors. Studies of this enzyme are inherently important, thus the knowledge gained from these structures lead to insights into metal ion specificity as well as inhibition. The overall content of the paper is sound, as well as the conclusions drawn from the structural data. Despite this, I have major reservations regarding the crystallographic data that is the basis for this manuscript. The Rmeas for all data sets are remarkably high, despite it being synchrotron data. The data needs to be re-processed and refined as their is no explanation given as to why this statistic is so poor. An Rpim needs to be given which can give a better understanding of the reliability of these data with such high Rmeas. For the two inhibitor complexes the story is even worse. The 9FXJ data is nonpunishable without re-processing. I would call this among the most egregious attempts i have observed to publish and refine poor quality data. The statistics for 9FXJ suggest the data is mostly uncorrelated. Despite ignoring the poor statistics, the data was still refined. No attention was paid to the 10% difference in refinement R values which clearly show the data was over refined. A 10% spread in R values, coupled to the 3.1 high resolution data bin Rmeas of 1.327 is highly outside of any normality. All four structures are overrefined. For the two inhibitor complexes, Polder omit maps must be shown.
Reviewer 2 Report
Comments and Suggestions for Authors
The manuscript “Metal ion binding to human Glutaminyl Cyclase: A structural Perspective” was written by Giusy Tassone and co-authored is based on crystallographic and in silico studies. The biological processes in Aβ accumulation are a contemporary area of research. Detailed knowledge of the aspects of interactions at the molecular level may allow us to explain the mechanisms of enzyme action and thus the methods of their regulation. This is the first stage leading to the development of new therapies, both preventive and therapeutic for dementia. The article is structured in a manner appropriate to the type of research presented and is written in a concise and clear style. The introduction is relatively brief and could be expanded upon. The discussion of the research results is justified and supported by the experimental evidence presented. Nevertheless, the article would benefit after including a few amendments. Please refer to the list below for a comprehensive overview of my comments. I would be grateful if you could address the points that I have highlighted, as I believe they will significantly enhance the quality of the manuscript.
1. The introduction is somewhat brief and does not provide a comprehensive rationale for further research. It is recommended that a paragraph be included in the text to elucidate the function of hQC in the context of neurodegenerative processes. It is well known that the hQC is considered an initiator agent for pathological Aβ accumulation. Inactive hQC can prevent amyloid accumulation in neurodegenerative diseases such as AD.
2. The fragment in lines 29-32 (“catalyze a post-translational chemical reaction in proteins or peptides that converts N-terminal glutamine or glutamate residues into N-terminal pyroglutamate (pE) by releasing ammonia or a water molecule, respectively.”) is the same as the sentence from article: Judite R.M. Coimbra, Paula I. Moreira, Armanda E. Santos, Jorge A.R. Salvador, Therapeutic potential of glutaminyl cyclases: Current status and emerging trends, Drug Discovery Today 28 (10), 2023, 103644, https://doi.org/10.1016/j.drudis.2023.103644
The authors do not even cite this work. I recommend rewording this fragment.
3. It is not evident how this work contributes to the existing body of knowledge regarding the activity of the enzyme in question. Does it affect the inhibition of action? What are the benefits of this work, and what are the intended applications of the resulting knowledge? Please expand on this aspect in the summary.
4. The authors concentrated on providing a comprehensive account of the interaction between metal ions, such as Co (II) and Zn (II) with hQC. However, there is no discernible correlation between this process and activity modulation. It is recommended that this section be marked.
5. It is recommended that Figure 7 be revised. In part A, the distance values are superimposed and therefore difficult to read. Furthermore, it would be beneficial to alter the color of the zinc atoms to enhance their visibility within the overall structure. At present, the color is identical to that of the carbon chain. Additionally, in parts B and C (Figure 7), zinc and amino acids should be more clearly visible, I suggest changing the font to black.
6. The Materials and Methods section requires more detail. All chemical reagents employed should be accompanied by the requisite information regarding their provenance, manufacturer, etc., in accordance with the stipulations of the MDPI publisher.
7. In my opinion, the lines 466-493 should be deleted.
8. Finally, I just have one more comment. It is interesting how interaction studies in crystal (solid) and in silico studies can be correlated with real enzymatic processes in the body at the cellular level. Can it be assumed that these interactions are the same?
Comments on the Quality of English LanguageThe quality of the English language is on a good level.
Reviewer 3 Report
Comments and Suggestions for Authors
This MS is about the crystal structures of hQC including metal and inhibitor complexes. Although it is not a ground-breaking work, it is of some interest to people in this field of study. The contents need a thorough check to eliminate mistakes including some typos, such as the extra “a sequence” in line 52, the misspelled “whit” in line 64, the interesting “olo” state in line 114 and the “Gli121” in line 252. For consistency, the one-letter codes of amino acids like N49 and N296 in line 57 as well as others in the introduction section should be replaced by three-letter codes. The chemical structures of PDB150 and SEN177 (or PDB-150, SEN-177) can be provided in Scheme 1 or in an additional scheme. The protein hQC is mentioned in abstract and introduction, but the crystal structures actually contain hQC-2X. What is hQC-2X? It should be clarified. Finally, judging by the large numbers of water molecules especially in the apo and Co(II) complexes (Table S1, or X1?), the crystal structures should have been extensively refined. However, the high Rfree values seem not indicating so. The authors are encouraged to re-examine the models to find out what problems are there.
Round 2
Reviewer 1 Report
Comments and Suggestions for Authors
The authors have addressed my concerns.